# Factors Influencing Patient Satisfaction and Loyalty as Perceived by Dentists and Their Patients

**DOI:** 10.3390/dj11090203

**Published:** 2023-08-28

**Authors:** Réka Magdolna Szabó, Norbert Buzás, Gábor Braunitzer, Michele Goldzieher Shedlin, Mark Ádám Antal

**Affiliations:** 1Department of Health Economics, Faculty of Medicine, University of Szeged, 6720 Szeged, Hungary; antal-szabo.reka@med.u-szeged.hu (R.M.S.); buzas.norbert@med.u-szeged.hu (N.B.); 2dicomLAB Dental, Ltd., 6726 Szeged, Hungary; braunitzer.gabor@gmail.com; 3Rory Meyers College of Nursing, New York University, New York, NY 10012, USA; mshedlin@aol.com; 4Department of Operative and Esthetic Dentistry, Faculty of Dentistry, University of Szeged, 6720 Szeged, Hungary

**Keywords:** loyalty, dental communication, dentist-patient communication, patient satisfaction

## Abstract

Objective: This study aimed to identify the key aspects of patients’ dental care experience that influenced their self-perceived satisfaction and loyalty. Also examined was the agreement between patients and dentists regarding these factors. Methods: Questionnaires were administered to 1121 patients and 77 dentists, focusing on demographic information and 15 selected items related to the patients’ last dental visit. Descriptive and linear regression analyses were conducted. Results: The study included participants from 41 practices. Factors significantly influencing satisfaction and loyalty included location convenience, treatment quality, trust in dentists’ decisions, visit frequency satisfaction, clear treatment explanations, dentist’s interest in symptoms, patient-dental personnel attachment, and dentist’s knowledge of the patient and their medical records. While overall agreement between patients and dentists was high, some areas exhibited notable disagreement. Conclusions: The findings mostly align with existing literature, underscoring the importance of communication, trust, and a personal patient-dentist relationship in promoting satisfaction and loyalty. However, they also show that local, generally not reported factors might be at play, which necessitates dentists’ awareness and consideration of the local context for optimal outcomes.

## 1. Introduction

Patient experience and practitioner communication as part of the overall patient experience are key factors of patient adherence [1,2], which means that these factors significantly contribute to therapeutic success or failure. It holds particularly true in the case of dentistry. While there exist numerous recognized medical phobias [3,4,5,6], odontophobia or dental fear ranks among the five most prevalent fears [7,8,9]. This interferes with attendance [10,11], and complete avoidance of the dentist may well be regarded as the ultimate therapeutic failure (with profound effects on the oral health of the public). Therefore, in dentistry, it is of utmost importance to provide a patient experience that leads to satisfaction, positive attitudes toward the dentist and the practice, and, in turn, a willingness to regularly attend. This has been frequently studied in the context of general practice [1,12,13,14,15,16], but less frequently regarding dentistry [17,18,19].

Patient satisfaction has been reported to be a multi-factorial phenomenon, with a complex set of objective and subjective elements [20]. Studies have reported that the quality of dentist-patient communication is related to patient satisfaction [20,21,22]. In the field of general medicine, studies have pointed out that patients prefer to be involved in the decision-making [23,24,25], and the few studies that are available on this specific question in dentistry, show the same [26,27]. It is also known that the perceived service quality influences patient loyalty through the effect of patient satisfaction which plays a key role in promoting patient loyalty [28].

Numerous studies have stressed the importance of communication in dentistry [29,30,31] and a few important conclusions have already been drawn regarding the success of dental communication. It has been shown that verbal communication in itself can influence patients’ satisfaction with treatment outcomes [32]. However, some studies indicate that dentists do not exploit this potential. For instance, Rozier and colleagues, in a large-sample, national-level survey, showed that US dentists utilized a narrow range of communication strategies and recommended more professional education in this area [33]. The quality of communication between the medical professional and the patient might be influenced by certain demographic and personal factors and concordances [34]. In a general medical setting, race concordance between the physician and the patient was found to result in longer visits characterized by more patient-positive affect [35]. Similar conclusions were drawn in connection with shared personal beliefs and values [36]. It is interesting that the gender of the dentist or physician also appears to be a key factor. Riley III et al. found that a male dentist was less likely to be aware of the importance of sharing information about the procedure to be performed than a female dentist [20]. The authors also reference Hall et al. [37], suggesting that healthcare providers may offer additional information and support to female patients. This is not necessarily due to assumptions about the health needs of women, but rather because female patients tend to openly express their feelings, concerns, questions, and preferences during discussions about medical choices. In addition, Thornton et al. suggest that sex concordance and age concordance can influence the quality of communication between the physician and the patient [38]. While the sex effect is well-known and studied [37,39,40], the effect of age and age discordance has been studied less often [17]. Several specific characteristics that determine good dentist-patient communication and lead to greater patient satisfaction have also been identified. To mention just a few: Most of the studies dealing with this topic found that a good explanation of the condition and its treatment is of utmost importance [16,20,41]. The treatment plan should be formulated through discussion and agreement [16]. It is similarly important that the dentist should explain what is going to happen before starting the procedure [41] and that the dentist should show interest when the patient is talking about his or her problems [16]. Finally, the importance of communication in dentistry is shown by the results of Lamprecht et al., who, in their study of patients’ criteria for choosing a dentist, found that dentists’ psychosocial skills appear to be the most important criteria for choosing a dentist [42].

Extra-communicative factors that may contribute to patient satisfaction and loyalty include trust in the physician’s judgments regarding one’s care and a good personal relationship between the patient and the care provider [12] or the patient’s level of knowledge about the healthcare services [43]. These factors and many others have been identified in different social and cultural settings, so they do not necessarily apply to any patient population.

An additional problem is that the opinions of the patient and the dentist regarding optimal and desirable dentist-patient communication may differ. Riley III et al. asked 197 dentists and their 5879 patients about patient satisfaction, seeking to identify concordance patterns [20]. Most of the patients were highly satisfied and the dentists correctly predicted this. However, among patients who were less than satisfied, there was a substantial subset of cases where the dentist was not aware of the patient’s dissatisfaction. It follows that to have a realistic picture of dentist-patient communication that can inform practice, the perspectives of both parties should be examined and compared [44,45,46].

In this study, our primary aim was to explore which aspects of the patients’ experience of their dental care influenced their self-perceived satisfaction and loyalty the most. To this end, we developed a questionnaire based on the literature [1,12,16,41,43,47], in which the patients were asked about their last visit to their dentist. A significant portion of the selected items characterized the patient experience and the communication of the dentist, but a few items raised general issues like the frequency, length, quality of the visits, satisfaction, and loyalty.

Our secondary aim was to examine the agreement between the patients’ experience and their dentists’ opinion about the importance of the same satisfaction- and loyalty-influencing aspects that the patients were asked about. This was achieved by a questionnaire created especially for dentists. This shorter questionnaire contained demographic items and 15 selected items that mirrored the corresponding items in the patient questionnaire, rephrased from the dentist’s perspective.

We formulated our hypotheses based on results published in the literature. Regarding the primary aim, we hypothesized that the dentist’s communication would be a major contributing factor to satisfaction and loyalty, especially clear language and good explanations. Regarding the secondary aim, we expected a generally high level of agreement over most of the items, with a few areas of disagreement.

## 2. Materials and Methods

### 2.1. Participants, Study Procedures, and Data Processing

Altogether 85 private dental practices from all over Hungary were contacted by email and invited to participate in this cross-sectional study. Our selection of practices for participation was not based on specific criteria; rather, we reached out to all 85 private dental practices for which we possessed contact information.

Sample size was calculated using G*Power 3.1 (Universität Düsseldorf, Düsseldorf, Germany). Assuming a multiple linear regression analysis, a significance level of *p* < 0.05, a medium effect size (f^2^ = 0.15), and the inclusion of 32 independent variables, the necessary sample size was projected to be *N* = 214. However, in the final analysis, we had a sample size of *N* = 1121, resulting in an achieved power of 1.0 (λ = 168.15, critical F = 1.45).

Of the contacted 85 practices, 41 agreed to participate. These practices were sent the electronic version of both the patient and the dentist questionnaires for printing and on-site administering, with instructions on how to administer the questionnaire. Consultation with the researchers was available at any time online or in person. In each practice, a dental assistant was tasked with administering the questionnaire to the participating dentists and their patients. The questionnaire was always administered in a quiet room, where the participant (either a dentist or a patient) was left alone to fill it in, after having received brief instructions that also appeared on the questionnaire itself in writing. Once all questionnaires in a practice had been filled in, they were sent back to the researchers who entered the responses into an Excel sheet. The responses were entered in a way that a patient’s response to a given item was always matched with that of his or her dentist so that agreement could be calculated between a dentist and his or her patients. When all questionnaires from all participating dentists had been received, the dataset was cleaned (data from dentists with less than 5 patients were removed, as well as the data of their patients), and the questions were coded for the blinded analysis. The coded datasheet was then sent to the independent evaluator for analysis (see Statistical analysis).

Participation was voluntary and anonymous for both the dentists and their patients. All dentists of all participating practices were invited to complete the questionnaire and to invite their patients to do so (i.e., recruitment took place on a self-selection basis). The inclusion and exclusion criteria for participant selection in this study encompassed active dental practitioners from the participating dental practices and patients who voluntarily agreed to participate, were native Hungarian speakers, and possessed the requisite cognitive capacity to comprehend the study’s objectives and questionnaire content. Exclusion criteria applied to individuals unable to provide informed consent or with limited cognitive ability to understand the study materials. Both the dentists and the patients signed an informed consent form. The informed consent forms were stored separately and did not contain any identifier that could allow making a connection between the questionnaires and the forms. The manager of the practice invited the dentists to participate, and those who agreed and filled in their questionnaire invited their patients. Patients and dentists were assigned a number on the site for the statistical analysis, but this number was never associated with any identifier. In this way, personal data were not processed in the study.

The study was approved by the Hungarian Medical Research Council’s Scientific and Research Ethical Committee (Approval number: IV/4834-2/2020/EKU).

### 2.2. The Questionnaires

#### 2.2.1. The Patient Questionnaires

First, the patient questionnaire of 31 items was developed (Appendix A). This contains six demographic items (Nos. 1 to 5 and 16), and the remaining 25 items were adapted from the literature on patient experience, satisfaction, loyalty, and practitioner-patient communication or used in their original form [1,12,16,41,43,47]. Adapting an item was necessary when it was originally used in the context of general medicine and worded accordingly. In these cases, the word “doctor” or “physician” was replaced with “dentist”. An example is item No. 12 (“I am very committed to continuing a relationship with my physician”) taken from Wang et al. [1], which appears as “I am very committed to continuing a relationship with my dentist” in our questionnaire.

Regarding satisfaction, we accepted the argument of Reichheld who proposed that the single most important measure of customer satisfaction is whether the customer would recommend a product or service to others [48]. As for loyalty, we accepted the definition by Oliver, who defines loyalty as “a deeply held commitment to re-buy or re-patronize a preferred product or service consistently in the future, despite situational influences and marketing efforts having the potential to cause switching behaviour” [49]. Two items specifically referred to satisfaction (I would recommend my dentist to others) and loyalty (I am very committed to continuing a relationship with my dentist). Because of their unambiguous phrasing and as they fit the concepts described in our definitions, we considered these the most suitable for the assessment of satisfaction and loyalty.

#### 2.2.2. The Dentist Questionnaires

Having decided on the questions to include in the patient questionnaire, we developed the dentist questionnaire of 19 items (Appendix A). Of these 19 items, 4 are demographic items, and 15 are the matched pairs of 15 items of the patient questionnaire. These 15 items were selected because they both covered important issues and were capable of being rephrased from the dentist’s perspective in a meaningful way so that the agreement between the opinion of the dentist and the experience of the patient could be assessed. In these cases, the dentists were asked about their opinion on a specific issue, and their patients were asked about the same issue regarding their last visit to their dentist. In this respect, we modeled our study on that of Riley and colleagues [20]. An example of such a matched pair is item No. 18 in the patient questionnaire (“The dentist was interested when I spoke about my symptoms.”) and item No. 9 in the dentist questionnaire (“It matters to the patients that their dentist shows interest when they speak about their symptoms.”). The aim of the dentist questionnaire was primarily to allow comparison with the patients’ perspective. Therefore, the dentist questionnaire contained fewer items than the patient questionnaire, where we considered a wider range of factors that could potentially influence satisfaction and loyalty. The dentist questionnaire was not developed as a standalone instrument, it was meant as a descriptive complement to the patient questionnaire.

All items in both questionnaires were 5-grade Likert-type statements, except for the demographic items, and one binary item where patients were asked to tell if they had visited their dentist more or less than 10 times by the time of the study (item No. 16).

#### 2.2.3. Pre-Testing and Psychometric Characteristics

Before administering the questionnaires to the study sample, a pilot test was conducted involving 25 dentists and 100 patients. The aim of this test was to assess the questionnaire’s reliability, internal consistency, and underlying factor structure. It’s important to note that the dentists and patients from the pilot sample were excluded from the final study sample.

For evaluating the factor structure of both questionnaires, an exploratory factor analysis (EFA) with principal component analysis was utilized, employing varimax rotation to ascertain item loadings within factors. Determining the number of factors to retain in the final model involved using the Kaiser factor retention method, assessing eigenvalues above 1, and employing a screen test. Item factor loadings were scrutinized, with a threshold of 0.50 used for item inclusion. As anticipated based on item selection, both questionnaires exhibited a clear two-factor arrangement. One factor pertained to the patient’s overall experience and personal rapport with the dentist, while the other encompassed communicative aspects like language usage. The study’s dependent variables aligned with the first factor. The Cronbach’s alpha values were calculated at 0.75 for the patient questionnaire and 0.79 for the dentist questionnaire.

Upon completion of the study sample dataset, a reevaluation of the questionnaires’ psychometric properties was undertaken. Both questionnaires maintained the same two underlying factors observed in the pilot phase. The patient questionnaire (*N* = 1121) exhibited the following characteristics in the final analysis: Bartlett’s test for sphericity yielded significance (χ^2^ = 10,544, df = 300, *p* < 0.01), the Kaiser-Meyer-Olkin (KMO) test for sampling adequacy produced an overall value of 0.904 (0.720–0.944), and Cronbach’s alpha was calculated at 0.84. Similarly, the dentist questionnaire analysis (*N* = 77) revealed significant results for Bartlett’s test (χ^2^ = 290, df = 120, *p* < 0.01), an overall KMO value of 0.608 (0.464–0.805), and a Cronbach’s alpha of 0.71. The somewhat lower values for the dentist questionnaire were to be expected, considering its auxiliary role as a descriptive complement to the patient questionnaire, rather than a standalone instrument.

The questionnaires were administered in Hungarian. To enable the use of the questionnaire with Hungarian patients and practitioners, the questionnaires were translated according to accepted international standards [50].

The questionnaires are attached as supporting documents in English with an indication of the sources of the items.

### 2.3. Statistical Analysis

#### 2.3.1. Descriptive Statistics and Hypothesis Tests

For the statistical analyses, SPSS 26.0 (IBM, Armonk, NY, USA) was used. For the descriptive characterization of the continuous variables, means, standard deviations, and the 95% confidence interval were used. The Likert-type responses were treated as continuous variables for all purposes as they represent degrees, not discrete choices. Categorical variables were described with frequencies. For hypothesis testing regarding the influencing factors of satisfaction and loyalty, regression analysis was used. In these regression models, Items 11 (regarding overall satisfaction) and 12 (regarding loyalty) from the patient questionnaire were the dependent variables, and the independent variables independent variables were the rest of the items. Note that items 25 to 28 in the patient questionnaire also explicitly refer to satisfaction, but in relation to specific aspects rather overall satisfaction, and were therefore used as independent variables. The practitioners’ demographic items (age, sex, location, and professional experience in years) were also included in these analyses. As the literature suggests that various demographic concordances between the practitioner and the patient (such as being of the same sex or being close in age) may influence the overall patient experience [36,51], we calculated three additional variables (location concordance, sex concordance, and age difference), and these were also added as independent variables.

#### 2.3.2. Dentist-Patient Comparisons

Agreement between dentists’ and patients’ responses was characterized in two ways. On the one hand, we determined which statements (items) the respondents agreed with the least and the most. This was done by calculating the 25th and 75th percentiles for the mean scores of all Likert-type items. Items scoring ≤the 25th percentile limit were considered the least agreed with and items scoring ≥the 75th percentile limit were considered the most agreed with. On the other hand, we introduced the variable “degree of disagreement” (DD), which was calculated for all 1121 dentist-patient response pairs, for all 15 matched item pairs, regardless of whether they appeared as significant factors in the satisfaction and loyalty analyses. DD was calculated as follows: if the patient’s score (PS) was lower than the dentist’s (DS), then PS was subtracted from DS and the result was multiplied by −1 (to express the direction of disagreement). On the contrary, if PS was higher than DS, then DS was simply subtracted from PS. This way, a negative value means that a certain item was given a higher score by the dentist, and a positive value indicates a higher score given by the patient. The value of full agreement is 0 and the value of full disagreement is either −4 or +4. Regardless of the sign, the higher the value, the higher the disagreement. At the level of an item pair, DD was expressed as the mean of all DD values for the given item pair, with SD and 95% CI. Besides DD, for each matched item pair, the percentages of dentist-patient responses in full agreement and full disagreement were also calculated.

## 3. Results

### 3.1. The Study Population

77 dentists and 1121 patients completed the questionnaire.

Of the responding dentists, 44 were male (51.9%) and 33 were female (48.1%). Their mean age was 40.57 (±15.23) years (23 to 72 years). By the time of the study, they had spent a mean of 17.60 (±12.16) years in the profession. Their practices were predominantly located in county seats (48 dentists, 62.3%), or other towns (19 dentists, 24.7%), and 10 practices were located in the capital (13.0%).

Among the respondents, 444 were male (39.6%) and 677 were female (60.4%). Their mean age was 43.60 (±13.97) years (18 to 90 years). Most of them lived in either a county seat (394 patients, 35.1%) or a town (425 patients, 37.9%). The rest of the patients lived in the capital (147 patients, 13.1%), in a township (12 patients, 1.1%), or a village (143 patients, 12.8%). Most of the patients had either a high school diploma (559 patients, 49.9%) or a university degree (509 patients, 45.4%). Twenty-two patients (2.0%) had a postgraduate degree (Ph.D.), while in the case of 31 patients (2.8%), finishing elementary studies was the highest level of education. 651 patients (58.1%), had visited their dentist less than 10 times by the time study, and the remaining 470 patients had had more than 10 visits.

The mean age difference between the dentists and their patients was 0.71 (±15.71) years, with a 95% confidence interval (CI) of −0.20 to 1.61 years (the values were negative when the patient was younger). The location of the dental practice and the patient’s residence matched in 706 cases (63.0%). The patient and the dentist were of the same gender in 598 cases (53.3%).

### 3.2. “I Would Recommend My Dentist to Others”—Satisfaction

The mean score of this statement among the patients was 4.92 (±0.31) with a 95% CI of 4.91–4.94. Among the dentists, the corresponding statement scored somewhat lower, 4.53 (±0.55) with a 95% CI of 4.41–4.66.

The results of the linear regression analysis indicated a significant contribution of the independent variables of the regression model to the overall variance of the patients’ responses (F (32,1088) = 27.59, *p* < 0.001, R^2^ = 0.43). Seven variables (questionnaire items) were found to significantly predict the score given to this statement. These were as follows: the match between the practice and the patient’s residence (β = −0.060, *p* < 0.05); the dentist expressed interest in the patient’s symptoms (β = 0.217, *p* < 0.001); the patient was content with the frequency of the appointments (β = 0.088, *p* < 0.01); the patient was content with the quality of the treatment (β = 0.146, *p* < 0.001); the patient felt that he or she could trust the dentist’s decisions about his or her treatment (β = 0.270, *p* < 0.001); the patient felt that the dentist knew him or her (β = 0.079, *p* < 0.05); and the patient felt that the dentist knew his or her medical records (β = 0.085, *p* < 0.01). In the case of 4 of the variables, it was possible to compare the patient’s experience and the dentist’s opinion on the given aspect of the patient-dentist relationship. The comparison of the mean scores is given in Table 1.

### 3.3. “I Am Very Committed to Continuing a Relationship with My Dentist”—Loyalty

The mean score of this statement among the patients was 4.78 (±0.71) with a 95% CI of 4.74–4.82. Among the dentists, the corresponding statement scored slightly lower, 4.45 (±0.62) with a 95% CI of 4.31–4.59.

The results of the linear regression analysis indicated a significant contribution of the independent variables of the regression model to the overall variance of the patients’ responses (F (32,1088) = 7.67, *p* < 0.001, R^2^ = 0.16). Six variables (questionnaire items) were found to significantly predict the score given to this statement. These were as follows: the dentist used clear language when explaining the treatment (β = 0.126, *p* < 0.01); the dentist explained what was going to happen before starting treatment (β = −0.101, *p* < 0.01); the patient was content with the frequency of the appointments (β = 0.125, *p* < 0.01); the dentist offered more than one treatment plans (β = 0.085, *p* < 0.05); the patient had the subjective feeling that the staff at his or her present dental care provider mattered to him or her (β = 0.098, *p* < 0.01); and the patient felt that he or she could trust the dentist’s decisions about his or her treatment (β = 0.091, *p* < 0.01). The dentist-patient comparison was possible in the case of 5 items. The comparison of the mean scores is given in Table 2.

### 3.4. Agreement/Disagreement between the Dentists’ and Their Patients’ Responses

The highest- and lowest-scoring statements (items) in both groups are summarized in Table 3. In the patient group, the limit of the 25th percentile was 4.42 and the limit of the 75th percentile was 4.9. In the dentist group, the limit of the 25th percentile was 4.05 and the limit of the 75th percentile was 4.77.

The results regarding the degree of disagreement are summarized in Table 4. The data in the table have been arranged according to the percentages of full agreement (i.e., when the dentist and the patient attributed the same level of importance to a given issue or found a statement to be true exactly to the same degree). This arrangement was chosen because this index is easy to interpret, and it reflects the general closeness of opinions very well.

The highest percentages of full agreement (>80%) were observed regarding the pairs D9-P26 (The dentist should show/showed interest in the patient’s symptoms) and D10-P28 (The dentist should explain/explained the problem with the teeth in an intelligible way). In contrast, the percentages of the full agreement were remarkably low (<30%) for three item pairs: D5-P8 (The patient trusts the dentist’s medical decisions according to the dentist/patient), D8-P14 (The patient is informed about healthcare according to the dentist/patient), and D15-P34 (The frequency of visits is important to the patient/satisfactory according to the patient). Relatively high rates of full disagreement were observed in the case of two item pairs: D7-P12 (The patient is strongly committed according to the dentist/patient) and D17-P37 (The dentist should offer/offered alternative treatment plan(s)).

The table shows that, even though the percentages of the full agreement were quite variable, the responses of the patients and dentists were still quite close. The mean disagreement was always below 1 point, except for the pair D15-P34 (The frequency of visits is important to the patient/satisfactory according to the patient), with a mean disagreement of 1.03 points. It is also clear from the table that most of the mean disagreement scores are positive, which indicates that the patients were somewhat more satisfied with the given aspect of patient experience than how important their dentist considered it. The two exceptions are D9-P26 (The dentist should show/showed interest in the patient’s symptoms) and D12-P31 (The duration of visits is important to the patient/satisfactory according to the patient), but the disagreement is marginal.

## 4. Discussion

In this study, our primary aim was to explore which aspects of the patients’ experiences of their dental care influenced their self-perceived satisfaction and loyalty the most. We also sought to compare patients’ and dentists’ perspectives on a variety of issues related to patient experience and the relationship between the patient and his or her dentist. All our hypotheses have been confirmed: good communication was indeed a major contributor to patient satisfaction and loyalty (with an emphasis on language use and good explanations) and there was generally a high degree of agreement between what the dentists considered important regarding the dental experience and what the patients expected. Yet, there were readily recognizable areas of disagreement.

In advance, it is important to stress that the concepts of satisfaction and loyalty are deeply intertwined, so when in our discussion we say that one factor influenced satisfaction and another one loyalty, it should not be understood exclusively. Rather, one had better understand it in a way that these factors contribute to the patients’ positive attitudes toward the dentist, expressed both as satisfaction and loyalty. The aim of this study was to get to know the patients’ and their dentist’s perspectives rather than to map all the complex relationships between the explanatory variables. Furthermore, it is important to stress that the model fit of our regression models is limited. This means there is a considerable amount of unexplained variance. In other words, this study identifies only a small portion of the factors that determine patient satisfaction and loyalty.

Studies from all over the world have dealt with patient satisfaction and loyalty, especially in connection with the field of general medicine [1,12,13,16], but a few dentistry-specific studies are also available [17,18,19,20,41]. To support our analysis of the results, we will be drawing upon these studies.

Our results are in line with the literature in at least two very important respects. The first one is the role of the trust [12,15]. The patients’ trust in their dentist’s decisions regarding their dental care turned out to be a significant predictor of both satisfaction and loyalty. As mentioned earlier, the two concepts are intertwined, and their exact relationship is a matter of discussion in the literature, especially regarding the role of trust. Platonova and co-workers argued that patient satisfaction bears patient loyalty, but at the same time, they are both significantly influenced by the trust [12]. In contrast, AlOmari and Hamid suggested that repeated satisfactory experience with the care provider builds trust, and the resulting trust leads to loyalty, so satisfaction comes first, and it builds loyalty through the trust [15]. In our opinion, this latter explanation is more plausible, while it cannot be excluded that established trust acts back on satisfaction (i.e., if one trusts one’s care provider, one will tend to be more satisfied with the care provider, in a self-fulfilling manner). In our patient population, trust in the dentist was generally high (the third highest-scoring item with a mean of 4.91 points). It is interesting that the dentists tended to underestimate their patients’ trust in them (a mean of 4.08 points), so much so that the matched item pair regarding trust turned out to be the one with the lowest percentage of full patient-dentist agreement (25.9%). It seems that the dentists had a somewhat pessimistic view of their patients in this respect, but the data gathered in this study do not allow an explanation. As we have found no other study to describe a similar phenomenon, we believe that this is a population-specific finding. In any way, based on our results, we agree with the literature that to establish patient loyalty, trust must be established first, which is best done by increasing patient satisfaction. What is the best approach to increasing satisfaction, which in turn leads to greater loyalty?

Regardless of the cultural context, it is an unequivocal finding that good communication on the care provider side is a major contributor to patient satisfaction and loyalty. This was found in Malta [18], the USA [20], Saudi Arabia [41], and the UK [19] alike, and our results from Hungary corroborate that observation. The apparently most important communicative strategy in dentistry in this respect is a thorough yet intelligible description of the procedures (especially directly before starting a procedure). This has an anxiety-reducing effect, which is of great importance in the dentistry [52]. Delivering professional explanations in simple language is also the first step to engage patients and make them partners in their own care, which has been shown to increase satisfaction [53]. The dentist in our sample seems to understand this well: all four highest-scoring items in the dentist group were related to good communication. That is, the dentists considered it especially important to show interest in their patients’ symptoms (4.94 points), use language that is easy to understand (4.88 points), explain the (dental) problem clearly (4.87 points), and discuss the treatment plan with their patients (4.77 points). Even more importantly, some of the same items showed up among the highest-scoring items of the patient group. It means that the patients did experience this patient-centered communication during their last visit. The patients reported that their dentist explained the (dental) problem clearly (4.91 points), they were satisfied with the explanation (4.91 points), their dentist showed interest in their symptoms (4.90 points), and their dentist told them what he or she was going to do before starting a specific procedure (4.90 points). In addition, the matched item pairs with the highest percentage of the full dentist-patient agreement were all related to communication/explanation, which means that the dentists did not only consider these issues highly important, but they could also meet their patient’s expectations at the same high level. It comes as no surprise that good communication appeared as a significant predictor of both satisfaction and loyalty. The dentist’s patient-perceived interest in the patient’s symptoms turned out to be a significant predictor of satisfaction, while explanations about one’s dental care in an “understandable” language, an explanation about the procedure that the dentist was about to start, and being offered more than one treatment plans had a significant effect on loyalty. Here we must point out again that the concepts of satisfaction and loyalty are intertwined, so these results are best understood as indicating that the named independent variables significantly contributed to a positive patient experience, which is reflected in a higher level of satisfaction and loyalty. All in all, these results corroborate our previous knowledge that it is of utmost importance that the language of the explanations regarding dental conditions and procedures be tailored to the patient and that explanations should be offered before all procedures.

Another question of interest is that of the personal relationship between the dentist and the patient. Little and colleagues [16] emphasized the importance of a personal relationship between the care provider and the patient. Platonova and co-workers also found that a good personal relationship with the care provider is important for patients to feel satisfied with the care provider. Our findings support this indeed: the subjective feeling of the patient that his or her dentist knew him or her contributed significantly to patient satisfaction, while a feeling of attachment and appreciation (“The people where I currently get my dental service matter to me”) contributed significantly to loyalty. The interesting finding regarding this is that these items scored low among the patients: three of the six items in the lowest percentile were related to the personal relationship with the dentist. Even if this does not mean that the relationship was perceived as definitely bad or weak (3.91–4.27 points of the five on average, depending on the particular item), these items did not rank even close to the communication items. That is, while the dentists were perceived as excellent communicators by their patients, their relationship with the dentist on a personal level was not seen by patients as a similarly remarkable aspect of the dental experience. We consider this a locally important finding. In Hungary (and probably in the entire post-Soviet bloc of Central Europe) the doctor-patient relationship is still often thought of in quite paternalistic terms [54]: the medical care provider is seen as an authority figure with whom connection at a personal level is inconceivable and whose sole task is to cure the physical ailment, to “fix” the patient, so to speak. In addition, Thompson and co-workers point out that medical paternalism is sometimes culturally accepted and expected [55]. Our experience with Hungarian patient populations is that the patients often think that it is rude to initiate a personal connection with the provider, while the providers often think that connecting to a patient is unwanted or even ethically risky. There is an awkward cautiousness on both sides, while both the literature and our results suggest that a good personal connection does contribute to the positive experience of the patient. We believe that putting more emphasis on this issue in courses of medical/dental communication and medical ethics (which all medical universities offer now) could be a good way to address this phenomenon in our geographical region.

A related issue is that of the dentist’s knowledge of the patient’s medical records. This had a significant effect on satisfaction. Of course, this has nothing to do with a dentist-patient relationship on a personal level, this is rather an indicator of professionalism and competence. The related item (“I’m confident that my dentist knows my medical records.”) ranked only 17 of 25 (4.65 points), which shows that the patients were not always completely convinced about this. Yet, on the one hand, a mean score of 4.65 of 5 is still quite high, and, on the other hand, this item is questionable in retrospect: unless the dentist makes it known to the patient in some explicit way that he or she knows the patient’s medical records, the patient has limited means to know about this, not to mention being confident. Some encounters do necessitate explicit signaling of such knowledge, but others do not. That is, this item measures the patient’s impression that is based on limited information. Thus, while we accept that the impression that the dentist knows about one’s medical data can foster satisfaction, we are reluctant to conclude that this result shows that the dentists in this sample were not aware of their patient’s medical data. All in all, the fact that this item ranked only 17th is probably due to its vague wording.

While it came as no surprise at all that the patient-perceived quality of the treatment had a significant effect on satisfaction, it was an unexpected finding that satisfaction with the frequency of the visits contributed significantly to both satisfaction and loyalty. To our knowledge, no previous study reported this. Lamprecht and colleagues do mention that the appointments should be convenient [42], but they do not mention frequency per se. From the results it seems that the dentists did not expect this result either: this item got the second lowest mean score in the dentist group (3.81 points), which means that the dentists did not consider this something highly important. The patients’ responses support this: satisfaction with the frequency of the visits ranked 15th among the patients, and this item was also characterized by the highest degree of patient-dentist disagreement and the third lowest percentage of full agreement (27.2%). Similarly to the findings of Riley III et al. [20], this aspect of dental care was a potential source of dissatisfaction for patients, yet dentists were not fully aware of its importance. While no study before has reported this specific issue, it is intuitive that if the frequency of the visits is tailored to the patient’s needs, the patient will be more satisfied. Therefore, we suggest that to increase patient satisfaction and loyalty, dentists should be aware of this phenomenon and put more emphasis on finding the optimal recall schedule for their patients.

Research suggests that female healthcare providers typically engage in longer consultations, provide more information, and express more explicit reassurance and encouragement compared to male clinicians [37,39,56]. In addition, studies have demonstrated that patients tend to adjust their responses based on the gender of their clinician, regardless of their gender [40,57]. Furthermore, the gender composition of the patient-provider relationship may impact overall patient satisfaction [58]. Based on this, we expected that the gender of the patient or the physician or their concordance would have a significant effect on the overall patient experience. Neither the patient’s nor the dentist’s gender nor the concordance of the two turned out to have a significant effect on either satisfaction or loyalty. It is difficult to give a good explanation for this, especially because the gender effect is reported in several studies from a variety of cultures and healthcare settings. In this respect, our results are not in line with the literature. While it is difficult to give a good explanation, it does not seem far-fetched to assume that the level of the personal connection between the dentist and the patient might play a mediating role here. While, to our knowledge, it has not been explicitly studied before, the results raise the possibility that the level of personal connection (which was suboptimal in our sample) acts as a permissive factor, and below a given level of personal connection, the effects of gender may not show (or to a much more limited extent). However, we would like to stress once again that this is just an assumption based on the results, we have no data or literary references to support this. At the same time, we consider this an assumption that would be worth testing from the standpoint of psychology.

As for the demographic factors, the single factor that had a significant effect on patient satisfaction was a match between the patient’s residence and the location of the dental office. We do not think that this has any special explanation other than that it is more convenient to visit a nearby dental office than one that is further away. Lamprecht and colleagues came to a similar conclusion [42].

Our findings provide support for the positive impact of effective communication, trust, and personalized rapport between patients and dentists on enhancing patient satisfaction and fostering loyalty. Our study has successfully reproduced the effects of well-documented factors, such as using patient-friendly language in professional explanations and the dentist’s explicit attention to patient symptoms. These recurring observations, transcending cultural boundaries, can be regarded as valuable guidelines for enhancing the patient experience. Yet, our results also underscore the significance of understanding local patient preferences. Notably, our study is the first to highlight the potential role of recall frequency in shaping the patient experience, suggesting a context-specific insight. This underscores the need to regularly assess patient experiences, solicit feedback on practice strengths and weaknesses, and create avenues for patients to contribute ideas for improvement. A succinct and anonymous patient satisfaction survey placed in a prominently visible location within the waiting room can serve as an effective means to collect such valuable insights.

In terms of future research directions, our study has left several questions unanswered and only scratched the surface in identifying factors that affect patient satisfaction and loyalty. The limited model fit in our regression models suggests that there are other factors at play. This limitation may be due to using questionnaire items that were already studied in the literature, which naturally narrowed down the scope of our research. Exploring patients’ own descriptions (through open-ended questions) of what contributes to their satisfaction with their dentist or influences their loyalty could be a productive avenue for further research.

With regards to the study’s strengths and limitations, there are several aspects to consider. The large size of the patient sample constitutes a pronounced strength, alongside the comprehensive incorporation of both patient and dentist perspectives. However, it is essential to acknowledge certain limitations. The study’s primary objective pertained to the exploration of patient and dentist viewpoints, rather than the establishment of intricate relationships among explanatory variables. While the patient sample is of a substantial size, its lack of representativeness across the broader Hungarian population must be recognized, a limitation further accentuated in the relatively modest dentist sample. Additionally, the sampling process inevitably introduced a measure of self-selection bias. Lastly, it is pertinent to underscore that the model fit of the regression models exhibits limitations, characterized by notable unexplained variance, a factor demanding careful consideration during result interpretation.

## 5. Conclusions

Most of the results of this study are in line with the published literature: the results corroborate that good communication, trust and a personal relationship between the patient and the dentist promote patient satisfaction and loyalty. We have managed to replicate the effect of widely reported factors like patient-friendly wording of professional explanations or the explicit interest of the dentist in the patient’s symptoms. Similarly to other studies, we have found that the perspectives of the dentist and the patient might differ on some key issues. It is important to know about these issues, as they offer points of intervention to improve the patient experience. For instance, optimizing the recall schedule to the individual patient’s needs is not difficult, but it appears that it might have a significant impact. At the same time, the results show that the well-known principles of patient satisfaction and loyalty can be modified by local factors. Therefore, we suggest that it is not enough to know the general principles, but to achieve the best possible outcome, the dentist should always be aware of and consider the preferences of the patient population they attend to.

## Figures and Tables

**Table 1 dentistry-11-00203-t001:** Comparison of the responses of the dentists and the patients regarding the significant factors of patient satisfaction. D: dentist questionnaire, P: patient questionnaire; the numbers next to the letters indicate the number of the item in the given questionnaire.

Item Pair		Topic	Dentist Mean (±SD)	Patient Mean (±SD)
D5	P8	The patient trusts the dentist’s medical decisions according to the dentist/patient	4.08 (±0.68)	4.91 (±0.33)
D9	P18	The dentist should show/showed interest in the patient’s symptoms	4.94 (±0.25)	4.90 (±0.38)
D15	P26	Frequency of appointments important to patient/satisfactory according to patient	3.81 (±0.90)	4.81 (±0.50)
D16	P27	The quality of treatment is important to the patient/satisfactory according to patient	4.70 (±0.59)	4.94 (±0.27)

**Table 2 dentistry-11-00203-t002:** Comparison of the responses of the dentists and the patients regarding the significant factors of patient loyalty. D: dentist questionnaire, P: patient questionnaire; the numbers next to the letters indicate the number of the item in the given questionnaire.

Item Pair		Topic	Dentist Mean (±SD)	Patient Mean (±SD)
D5	P8	The patient trusts the dentist’s medical decisions according to the dentist/patient	4.08 (±0.68)	4.91 (±0.33)
D12	P23	The dentist should use/used clear language when talking about the treatment	4.88 (±0.32)	4.83 (±0.46)
D13	P24	The dentist should explain/explained the procedure before starting	4.66 (±0.50)	4.90 (±0.34)
D15	P26	The frequency of appointments important to the patient/satisfactory according to the patient	3.81 (±0.90)	4.81 (±0.50)
D17	P29	The dentist should offer/offered alternative treatment plan(s)	4.44 (±0.87)	4.57 (±0.80)

**Table 3 dentistry-11-00203-t003:** The highest- and lowest-scoring statements about patient experience according to the patients and their dentists. A higher score indicates a higher level of agreement with the statement. The statements are arranged in descending order by their mean score.

PATIENTS			
Item No.	Item	Mean Score	
27	I am satisfied with the quality of the treatment.	4.94	75th percentile
11	I would recommend my dentist to others.	4.92	
8	I trust this dentist’s judgments about my medical care.	4.91	
20	The dentist explained clearly what the problem was.	4.91	
28	I am satisfied with the explanation given by the dentist.	4.91	
18	The dentist was interested when I spoke about my symptoms.	4.90	
24	The dentist told me what s/he was going to do before starting the procedure.	4.90	
7	I have developed a personal relationship with my current dentist	4.27	25th percentile
9	I’m confident that my dentist knows me.	4.22	
14	I possess good knowledge of health care services.	4.00	
19	The dentist was interested in the effects of the problem on my family or private life.	3.91	
13	For me, the costs in time/money/effort to switch dentists are high.	3.59	
15	I am quite experienced in the health care area.	3.47	
DENTISTS			
9	It matters to the patients that their dentist shows interest when they speak about their symptoms.	4.94	75th percentile
12	It matters to the patients that their dentist uses words that are understandable in talking about their dental care.	4.88	
10	It matters to the patients that their dentist explains clearly what the problem is.	4.87	
18	It matters to the patients that their dentist discusses the treatment plan with them.	4.77	
12	It matters to the patients that the dentist encourages them to ask questions about their treatment.	4.05	25th percentile
13	The duration of an appointment matters to the patients.	4.05	
14	The frequency of appointments matters to the patients.	3.81	
15	Most of the patients possess good knowledge of health care services.	3.48	

**Table 4 dentistry-11-00203-t004:** Agreement/disagreement between the dentists’ and their patients’ responses to all matched item pairs arranged in ascending order of full agreement. Results from 1121 matched responses. The numbering of the items follows the convention of Table 1 and Table 2. For the calculation of Degree of Disagreement, see the Statistical Analysis section.

Item Pair		Topic	Full Agreement N (%)	Full Disagreement N (%)	Degree of Disagreement (Mean)	SD	95% CI Lower Limit	95% CI Upper Limit
D5	P8	The patient trusts the dentist’s medical decisions according to the dentist/patient	291 (25.9%)	0 (0%)	0.90	0.80	0.85	0.95
D8	P14	The patient is informed about healthcare according to the dentist/patient	300 (26.7%)	4 (0.4%)	0.48	1.23	0.41	0.55
D15	P34	The frequency of visits is important to the patient/satisfactory according to the patient	305 (27.2%)	0 (0%)	1.03	1.02	0.97	1.09
D14	P33	The duration of visits is important to the patient/satisfactory according to the patient	398 (35.5%)	0 (0%)	0.86	0.86	0.80	0.92
D11	P30	The dentist should encourage/encouraged questions about treatment	407 (36.3%)	0 (0%)	0.59	0.57	0.53	0.66
D6	P11	The patient would recommend the dentist to others according to the dentist/patient	570 (50.8%)	0 (0%)	0.41	0.64	0.37	0.44
D7	P12	The patient is strongly committed according to the dentist/patient	582 (51.9%)	14 (1.2%)	0.24	0.96	0.18	0.29
D17	P37	The dentist should offer/offered alternative treatment plan(s)	590 (52.6%)	17 (1.5%)	0.04	1.11	−0.02	0.11
D19	P39	The dentist and the patient should agree/agreed on the treatment plan	722 (64.3%)	2(0.2%)	0.23	0.66	0.19	0.27
D13	P32	The dentist should explain/explained the procedure before starting	728 (64.9%)	0 (0%)	0.26	0.55	0.23	0.30
D16	P35	The quality of treatment is important to the patient/satisfactory according to the patient	770 (68.6%)	0 (0%)	0.31	0.77	0.27	0.36
D18	P38	The dentist should discuss/discussed treatment plan with the patient	824 (73.4%)	0 (0%)	0.14	0.59	0.11	0.17
D12	P31	The dentist should use/used clear language when talking about the treatment	863 (76.9%)	0 (0%)	−0.06	0.57	−0.09	−0.02
D10	P28	The dentist should explain/explained the problem with the teeth in an intelligible way	935 (83.3%)	1 (0.1%)	0.03	0.47	0.00	0.05
D9	P26	The dentist should show/showed interest in the patient’s symptoms	958 (85.4%)	1 (0.1%)	−0.03	0.47	−0.05	0.00

## Data Availability

The analysis dataset is available from the corresponding author on reasonable request.

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
