# Peer review of "Factors Influencing Patient Satisfaction and Loyalty as Perceived by Dentists and Their Patients"

_dentistry, 2023, doi:10.3390/dj11090203_

Round 1

Reviewer 1 Report

I consider this report to be very successful. The authors address the current state of knowledge and manage to reflect and discuss their own results in a good context. There are some small errors in the text, these should be corrected.

Please check:

- Lines 62,89,...: Reference named correct? Riley III...

Please check:

- Sentence from L. 64-68: editing needed ?

- L. 407: important in dentistry

- L. 408: ... step to engage (?)

Author Response

Dear Reviewer,

Thanks for your comments. Please find our detailed responses below:

I consider this report to be very successful. The authors address the current state of knowledge and manage to reflect and discuss their own results in a good context. There are some small errors in the text, these should be corrected.

Please check:

- Lines 62,89,...: Reference named correct? Riley III...

R: This is the way the author uses his name in all his publications (sometimes also Joseph L. Riley, 3rd.). It would be probably disrespectful to delete the number.  

Comments on the Quality of English Language:

Please check:

- Sentence from L. 64-68: editing needed ?

R: Indeed, the sentence was a bit lengthy, so we broke it down into two sentences to make the argument easier to follow.

- L. 407: important in dentistry

R: Corrected, thanks.

- L. 408: ... step to engage (?)

R: Missing infinitive particle added, thanks.

Reviewer 2 Report

Line 33-35: this sentence is simply inaccurate and sounds too dentistry-focused. Dentistry as a surgical profession is perceived indifferently from other surgical specialities, which are associated with tomophobia (10.1186/1752-1947-3-131), iatrophobia (10.1016/j.pec.2019.06.006), trypanophobia (10.1016/j.cegh.2023.101257), and white coat syndrome (10.2147/IBPC.S152761). This whole statement has to be rewritten reasonably without an over-emphasis on dentophobia.

Lines 96-124: these three paragraphs are unnecessarily long. Most of their content should be moved to the Methods section. One concise paragraph at the end of the Introduction should explicitly state the study's primary and secondary objectives.

The methods section should be restructured according to the STROBE checklist for cross-sectional studies.

Line 127-129: which recruitment method was used? Random or non-random?

Inclusion & exclusion criteria should be stated explicitly.

Sample size calculation should be stated explicitly.

Regarding the instrument, it is unclear why the authors decided to do a validated translation from English to Hungarian. Forward and backward translations are recommended only when the instrument has to be equivalent to enable cross-cultural comparisons. In this study, you had only participants from Hungary; you could have developed a Hungarian version of the questionnaire directly. 

Regarding the instrument, there is no information about the psychometric properties of the questionnaire or how it was pre-tested.

Blinding of the statistician does not make much sense in the context of this study, as this is not a clinical trial or sensitive experimental study. If the authors do not want to credit the person who analysed the data as a co-author, then this is another problematic issue that can't be addressed in this way. Hiring a biostatistician to analyse your data and paying them financial compensation for their work can not disqualify them as co-authors.

Line 211: the term "predictors" is inappropriate to be used in a cross-sectional survey-based study. You can use correlates.

Regarding the statistical analysis, the section is too long and can be divided into sub-sections, or some of its content can be placed under subheadings "variables" or "outcome measures".

Regarding the regression analysis, the assumptions of linear regression (e.g., linearity, homoscedasticity, independence, normality) might not be met. That should be clarified.

Treating the Likert-like items as continuous variables might not be appropriate as these responses represent ordinal data. They should at least be normally distributed. Normality tests were not reported.

Regarding data filtering, the study mentioned that data from dentists with less than 5 patients were removed. However, it did not provide a clear justification for this decision.

Tables headings are too long. They should be descriptive and concise.

Tables footers should highlight the types of inferential tests used in each table.

The model fit of linear regression models, especially the loyalty one, is too limited. The R-squared value of 16% is poor. You should amend your results accordingly and their interpretation.

The wording of items needs to be revised; some items just do not make sense, for example, "The patient trusts the dentist’s medical decisions according to the dentist/patient", "The dentist should use/used clear language when talking about the treatment".

The Discussion section does not provide implications for future research or current practice.

The Discussion section should provide a clear paragraph on the strengths and added value of this study.

The manuscript is written in an acceptable language. Only the wording of items need to be revised.

Author Response

Dear Reviewer,

Thanks for your comments. Please find our detailed responses below:

Line 33-35: this sentence is simply inaccurate and sounds too dentistry-focused. Dentistry as a surgical profession is perceived indifferently from other surgical specialities, which are associated with tomophobia (10.1186/1752-1947-3-131), iatrophobia (10.1016/j.pec.2019.06.006), trypanophobia (10.1016/j.cegh.2023.101257), and white coat syndrome (10.2147/IBPC.S152761). This whole statement has to be rewritten reasonably without an over-emphasis on dentophobia.

R: We value and acknowledge your valuable insights concerning the discussion of medical phobias in our manuscript. It is essential to clarify that our intention was not to overlook the existence of diverse medical phobias associated with healthcare practitioners in general, medical procedures, or medical equipment. Rather, our objective was to highlight dentistry's distinctive status as a medical specialization characterized by its own unique phobia, in contrast to fields such as cardiology, gynecology, orthopedics, and others. While our assertion regarding dental fear's prominence among the top five prevalent fears remains supported, your feedback prompted us to revise the statement, ensuring clarity and accuracy. As per your suggestion, we have rephrased the sentence and included the recommended references to enhance the manuscript.

Lines 96-124: these three paragraphs are unnecessarily long. Most of their content should be moved to the Methods section. One concise paragraph at the end of the Introduction should explicitly state the study's primary and secondary objectives.

R: The reviewer is right; a significant portion of this section predominantly focused on methodology. We have revised the section and reorganized the manuscript. As part of the reorganization, we have relocated the methodological components to the dedicated methodological section of the paper.

The methods section should be restructured according to the STROBE checklist for cross-sectional studies

R: We appreciate your notification regarding the journal's expectation to align with STROBE guidelines in organizing the Methods section. To rectify the identified gaps, we have incorporated the necessary details, including the recruitment method, inclusion/exclusion criteria, and sample size calculation. Taking further measures, we have meticulously reviewed the STROBE checklist again to confirm that our Methods section now comprehensively encompasses all the stipulated elements.

Line 127-129: which recruitment method was used? Random or non-random?

R: Recruitment was conducted through voluntary participation, as indicated in the revised version of the manuscript. While we acknowledge that this approach introduces inherent self-selection bias, it is important to adhere to ethical principles outlined in the Declaration of Helsinki, where participants willingly engage in medical studies. This intrinsic self-selection bias is inherent in ethically conducted studies involving human subjects, and while random sampling may be technically feasible, it would still entail drawing samples from a pool of self-selected individuals. Nonetheless, it is important to note that our selection of practices for participation was not based on specific criteria; rather, we reached out to all 85 private dental practices for which we possessed contact information.

Inclusion & exclusion criteria should be stated explicitly.

R: This information has been added as follows:

All dentists of all participating practices were invited to complete the questionnaire and to invite their patients to do so (i.e., recruitment took place on a self-selection basis). The inclusion and exclusion criteria for participant selection in this study encompassed active dental practitioners from the participating dental practices and patients who voluntarily agreed to participate, were native Hungarian speakers, and possessed the requisite cognitive capacity to comprehend the study's objectives and questionnaire content. Exclusion criteria applied to individuals unable to provide informed consent or with limited cognitive ability to understand the study materials.

Sample size calculation should be stated explicitly.

R: Sample size was calculated using G*Power 3.1 (Universität Düsseldorf, Germany). Assuming a multiple linear regression analysis, a significance level of p<0.05, a medium effect size (f2=0.15), and the inclusion of 32 independent variables, the necessary sample size was projected to be N=214. However, in the final analysis, we utilized a sample size of N=1121, resulting in an achieved power of 1.0 (λ=168.15, critical F=1.45).

Regarding the instrument, it is unclear why the authors decided to do a validated translation from English to Hungarian. Forward and backward translations are recommended only when the instrument has to be equivalent to enable cross-cultural comparisons. In this study, you had only participants from Hungary; you could have developed a Hungarian version of the questionnaire directly. 

R: Thank you for your feedback regarding our translation process. We appreciate your consideration of the approach we took. While we acknowledge that the study participants were solely from Hungary, our decision to undertake a validated translation from English to Hungarian was driven by our commitment to ensuring the highest quality of translation. We aimed to uphold the integrity of the instrument's content and minimize the potential for any linguistic nuances to affect the accuracy of participant responses. However, the Reviewer is right, this is not crucial information; the details of the translation-backtranslation process have been deleted.

Regarding the instrument, there is no information about the psychometric properties of the questionnaire or how it was pre-tested.

R: We appreciate your suggestion to incorporate pre-testing and psychometric aspects into the manuscript. Initially, we omitted these aspects as we had analyzed the data at the level of individual variables instead of utilizing the factor structure for this purpose. Additionally, the manuscript's length was a consideration. However, upon reflection, we recognize the importance of including this information. As a result, we have now included the most important results of both the pilot analyses and the final outcomes.

Blinding of the statistician does not make much sense in the context of this study, as this is not a clinical trial or sensitive experimental study. If the authors do not want to credit the person who analysed the data as a co-author, then this is another problematic issue that can't be addressed in this way. Hiring a biostatistician to analyse your data and paying them financial compensation for their work can not disqualify them as co-authors.

R:  We sincerely appreciate your insights and concerns regarding the blinding of the statistician in the context of our study. We acknowledge that blinding may not be conventionally applied in all types of research; however, it has been a customary practice in our institution to ensure unbiased data analysis. The intention behind blinding was solely to maintain the integrity of the analysis process. At the same time, we agree that this is an unnecessary detail that just made the analysis section longer.

We want to clarify that our decision to not credit the statistician as a co-author was not meant to undermine their contributions or the importance of his role. The decision was mutually agreed upon, and the statistician was explicitly acknowledged in the Acknowledgements section for his valuable contributions.

In light of your feedback, we reevaluated this aspect and had a discussion with the statistician. We are pleased to inform you that, based on your suggestion, the statistician has been added to the author list to duly recognize his contributions.

Line 211: the term "predictors" is inappropriate to be used in a cross-sectional survey-based study. You can use correlates.

R: In the context of line 211, we meant independent variables. The term has been corrected accordingly.

Regarding the statistical analysis, the section is too long and can be divided into sub-sections, or some of its content can be placed under subheadings "variables" or "outcome measures".

R: Two subheadings have been added. Subheadings have also been added to the section describing the questionnaires, which has become quite long after the addition of the part on pre-testing and psychometric charactersitics.   

Regarding the regression analysis, the assumptions of linear regression (e.g., linearity, homoscedasticity, independence, normality) might not be met. That should be clarified.Treating the Likert-like items as continuous variables might not be appropriate as these responses represent ordinal data. They should at least be normally distributed. Normality tests were not reported.

R:  This clarification has been included in the manuscript. It is worth noting that the treatment of Likert-like items as continuous variables is a commonly adopted practice (as demonstrated, for example, in this source: https://www.ncbi.nlm.nih.gov/pmc/articles/PMC8623523/). While this approach remains a subject of ongoing discussion, certain scholars argue in favor of treating such variables as continuous in a majority of cases (as elaborated in this publication: https://www.frontiersin.org/articles/10.3389/feduc.2020.589965/full ).

Regarding data filtering, the study mentioned that data from dentists with less than 5 patients were removed. However, it did not provide a clear justification for this decision.

R: This decision was made on a discretionary basis due to the occurrence of situations where the opinions of only one or two patients could be juxtaposed with those of the dentist. In our opinion, such scenarios do not allow a serious comparison, even if the decision is only descriptive.

Tables headings are too long. They should be descriptive and concise.

R: We have shortened them, especially where the extra information (making the heading long) was available in the main text too. An example is Table 4.

Tables footers should highlight the types of inferential tests used in each table.

R: The manuscript contains no table in which the results of inferential tests are shown. The tables show descriptive statistics only.

The model fit of linear regression models, especially the loyalty one, is too limited. The R-squared value of 16% is poor. You should amend your results accordingly and their interpretation.

R: Indeed, this is clearly a limitation that we failed to emphasize enough. We have amended the manuscript accordingly.

The wording of items needs to be revised; some items just do not make sense, for example, "The patient trusts the dentist’s medical decisions according to the dentist/patient", "The dentist should use/used clear language when talking about the treatment".

R: We acknowledge your observation; however, the instances you referred to do not pertain to the actual items per se. The utilization of such phrasing is intended to denote item pairs that address the same issue but from distinct perspectives. This distinction is reflected in the tables, where the header designates "item pair" rather than solely "item." It is noteworthy that Table 3, for example, presents the items in their precise wording. Additionally, a comprehensive listing of items is available for reference in Appendices A and B.

The Discussion section does not provide implications for future research or current practice.

R: The sections has been added.

The Discussion section should provide a clear paragraph on the strengths and added value of this study.

R: This has been added to the end of the last paragraph.

Comments on the Quality of English Language The manuscript is written in an acceptable language. Only the wording of items need to be revised.

R: Thanks, regarding the wording of the items, see our comment above.

Reviewer 3 Report

I found this manuscript well written and of great interest to a practitioner.    The study was well designed and the authors very carefully analyzed their data in the context of existing literature in the area. Its only minor flaw is it was perhaps more wordy than necessary.   I think it will certainly be cited in the future by others looking at the relationships between dental practitioners and their patients.  Similar studies regarding dentists and the parents of dental patients would be of great interest to the reviewer.  

Author Response

Dear Reviewer,

Thanks for your comments. Please find our detailed responses below:

I found this manuscript well written and of great interest to a practitioner.    The study was well designed and the authors very carefully analyzed their data in the context of existing literature in the area. Its only minor flaw is it was perhaps more wordy than necessary.   I think it will certainly be cited in the future by others looking at the relationships between dental practitioners and their patients.  Similar studies regarding dentists and the parents of dental patients would be of great interest to the reviewer.  

R: We sincerely appreciate the time and effort you dedicated to reviewing our manuscript. Thank you.

Round 2

Reviewer 2 Report

Thank you for addressing each of my earlier comments properly and satisfactorily. The manuscript has been improved significantly.